# Toward Reliable Domain Adaptation under Continuous Spurious Shift

## Abstract

Recent advances in domain adaptation have shown promise in transferring knowledge across domains characterized by a continuous value or vector, such as varying patient ages, where "age" serves as a continuous index. However, these approaches often fail when spurious features shift continuously along with the domain index. This paper introduces the first method designed to withstand the continuous shifting of spurious features during domain adaptation. Our method enhances domain adaptation performance by aligning causally transportable encodings across continuously indexed domains. Theoretical analysis demonstrates that our approach more effectively ensures causal transportability across different domains. Empirical results, from both semi-synthetic and real-world medical datasets, indicate that our method outperforms state-of-the-art domain adaptation methods.

## 1 Introduction

Machine learning typically presumes that the training and test data comes from identical distributions, hoping that the trained model effectively generalizes to the test environment. However, this presumption breaks when the training and testing occur across different domains (e.g., with distinct source and target domains). Domain adaptation (DA) effectively addresses this challenge by utilizing labeled data from the source domain along with either unlabeled or minimally labeled data from the target domain, thereby improving model performance (Ben-David et al., 2010; Ganin et al., 2016; Tzeng et al., 2017; Zhang et al., 2019).

Continuously indexed domain adaptation (CIDA) (Wang et al., 2020) generalizes typical DA, which focuses on discrete domains (transferring from dataset A to B), to DA across continuously indexed domains, where domain shift is characterized by a continuous index such as time and location. For instance, in healthcare, CIDA can be instrumental in transferring knowledge across patient data that varies with age. As patients age, their physiological parameters and response to treatments can change subtly; in response, CIDA aims to train a model that can adapt to these continuous shifts. Unfortunately, previous DA methods (Ganin et al., 2016; Tzeng et al., 2017; Zhang et al., 2019), including CIDA (Wang et al., 2020), often fail when spurious features are continuously shifting.

**Example 1 (Continuously Shifting Spurious Features in Sleep Studies).** *Suppose one trains a model to take as input a time series of breathing signal* $\mathbf{x} \in \mathbb{R}^T$ *to predict the corresponding sleep stage as* $y^{pred} \in \{$*'Awake', 'Light Sleep', 'Deep Sleep', 'REM Sleep'*$\}$*. Here 'respiratory rate' is a typical spurious feature. If one trains the model using data from young subjects (with age* $20 \sim 30$*), it will learn that 'higher respiratory rates' often correspond to 'REM Sleep'. However, as subjects continuously age, they may have a slower respiratory rate in breathing patterns. Therefore the spurious feature 'respiratory rate' no longer works when predicting sleep stages for older subjects, and the older the subject is, the lower the model accuracy.*

Our analysis shows that either alignment-based methods (Ganin et al., 2016; Wang et al., 2020) or causality-inspired methods (Mao et al., 2022) alone do not solve the problem. Alignment-based methods tend to align spurious features rather than causal features, therefore often fail to generalize across continuously indexed domains with continuously shifting spurious features. On the other hand, causality-inspired methods aiming to learn causal features (encodings) may collapse to non-causal (association-based) methods (see Sec. 3 for detailed analysis), therefore also failing to generalize. Motivated by such analysis, we then propose to jointly (1) infer causally transportable

Table 1: Comparison of our CADA with different representative previous methods.

| | CUA | ADDA | DANN | CDANN | MDD | CIDA | VOOD | GDA | CADA (Ours) |
|---|---|---|---|---|---|---|---|---|---|
| Continuous | ✗ | ✗ | ✗ | ✗ | ✗ | ✓ | ✗ | ✓ | ✓ |
| Causal | ✗ | ✗ | ✗ | ✗ | ✗ | ✗ | ✓ | ✗ | ✓ |
| Multi-Domain | ✓ | ✗ | ✓ | ✓ | ✗ | ✓ | ✗ | ✓ | ✓ |
| Covariate Shift | ✓ | ✓ | ✓ | ✓ | ✓ | ✓ | ✗ | ✓ | ✓ |
| Setting (DA/DG) | DA | DA | DA | DA | DA | DA | DG | DA | DA |

encodings (representations) and (2) align these encodings across continuously indexed domains, thereby improving domain adaptation performance under continuous spurious shift. Our contributions are as follows:

- We identify the problem of continuous spurious shift and propose Continuously trAnsportable Domain Adaptation (CADA) as the first general DA method to address this problem.
- Our theoretical analysis shows that our CADA can better ensure causal transportability across continuously indexed domains.
- Empirical results on both semi-synthetic and real-world medical datasets show that our method outperforms the state-of-the-art DA methods in the face of continuous spurious shift.

## 2 RELATED WORK

**Typical Domain Adaptation.** Domain Adaptation has long been studied (Farahani et al., 2021; Csurka, 2017; Ben-David et al., 2010; Peng et al., 2019; Prabhu et al., 2021; Liu et al., 2023; Xu et al., 2022) to promote model's generalization ability on unseen domains, with unlabeled or a limited amount of labeled data. Traditional methods include importance weighting (Shimodaira, 2000; Gretton et al., 2009; Lipton et al., 2018), self-training (Zou et al., 2018; Kumar et al., 2020; Prabhu et al., 2021), distribution matching (Pan et al., 2010; Tzeng et al., 2014; Sun & Saenko, 2016; Peng et al., 2019; Nguyen-Meidine et al., 2021; He et al., 2024) and adversarial-based training (Ganin et al., 2016; Zhang et al., 2019; Zhao et al., 2017; Wang et al., 2020; Xu et al., 2023). These methods either (1) only focus on categorical domains, failing to handle continuous domains or (2) struggles to eliminate the influence of continuously shifting spurious features. In contrast, our method successfully addresses these two major challenges (see empirical results in Sec. 5).

**Causality-Inspired Domain Adaptation.** Causal inference is a powerful method for modeling knowledge transfer (Bareinboim & Pearl, 2014; 2013; 2016; Bühlmann, 2020; Correa & Bareinboim, 2019; 2020; Magliacane et al., 2018; Rojas-Carulla et al., 2018) and ensuring structure invariance (Pearl et al., 2000). Various studies have explored its application to eliminate spurious features and improve domain adaptation performance (Arjovsky et al., 2019; Mahajan et al., 2021; Mao et al., 2021; Yue et al., 2021). For instance, VOOD (Mao et al., 2022) aims to estimate invariant causal effect across different environments, hoping to improve domain generalization (DG) performance. However, these causal methods (1) are still subject to covariate shift, often compromising the causal model's identifiability (Pearl et al., 2000), (2) can easily degenerate to association-based method, and (3) are not designed for continuously indexed domains. In contrast, our CADA effectively addresses these problem via joint latent-space distribution alignment and causal inference.

**Comparison of Representative Methods.** Table 1 summarizes the differences between our CADA and representative previous methods in terms of whether they (1) are designed for continuously indexed domains, (2) are causal/robust against spurious features, (3) can naturally handle multiple source/target domains, (4) are robust against covariate shift, and (5) are designed for DA or DG.

## 3 THEORETICAL ANALYSIS

In this section, we describe the challenges of performing domain adaptation on data with continuously shifting spurious correlations (our solution in Sec. 4). **All proofs are available in Appendix A.**

**Notation and Problem.** We consider the problem of unsupervised continuously indexed domain adaptation as proposed in (Wang et al., 2020). We assume that the continuous domain index set $\mathcal{K}$ is a part of a metric space, $\mathcal{K} = \mathcal{K}_s \cup \mathcal{K}_t$, with $\mathcal{K}_s$ and $\mathcal{K}_t$ as domain index sets for the source and target domains, respectively. The input and labels are denoted as $\mathbf{x}$ and $y$, respectively. Given the labeled

source-domain data $(\mathbf{x}_i^s, y_i^s, k_i^s)_{i=1}^n$ and unlabeled target-domain data $(\mathbf{x}_i^t, k_i^t)_{i=1}^l$, with $k_i^s \in \mathcal{K}_s$ and $k_i^t \in \mathcal{K}_t$, the goal is to predict the target-domain labels $(y_i^t)_{i=1}^l$. (Note that the labels $(y_i^t)_{i=1}^l$ are only available at test time for evaluation purpose only.) We use upper-case letters (e.g., $X$) to denote random variables and lower-case letters (e.g., $x$) to denote corresponding realizations.

## 3.1 A Causal View on Continuously Indexed Domain Adaptation

**Structural Causal Model.** We use the Structural Causal Model (SCM) in Fig. 1 to describe the underlying generative process of the random variables the data $X$ and their labels $Y$. In addition to the observed variables $X$ and $Y$, this SCM also involves the following variables:

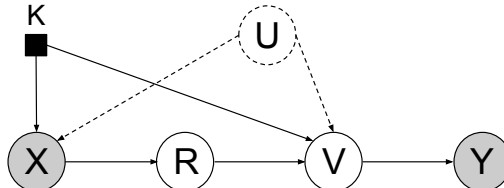

- $R$, which is a vector encoding both causal and spurious features of the input data $X$,
- $V$, which is the input's causal factors extracted from the representation $R$,
- $U$, which is the unobserved noise and spurious features from external sources, and
- $K$, which is the continuous domain index.

Figure 1: Causal diagram for CADA. Shaded and transparent circles denote observed and latent variables, respectively. Dotted circle denotes an unobserved confounder. Dotted arrows denote unobserved dependence from the confounder.

$U$ and $K$ together affect both the generation of input $X$ and the extraction of causal factors $V$ from $R$; in other words, $U$ and $K$ guide the process of eliminating spurious features from $R$. Note that in the context of causal inference, $K$ can be considered as a continuous version of the *transportability node* (Pearl & Bareinboim, 2011); it points to variables whose generation process differs for different domains.

Following Example 1, below we provide an example of using the SCM in Fig. 1 for sleep studies across continuously indexed domains (patients of different ages).

**Example 2** (**Causal Model for Sleep Studies**). *In sleep studies (Zhao et al., 2017), one typical task is to estimate the sleep stage $Y$ (e.g., "Awake" and "Deep Sleep") from a patient's breathing signal $X$; here "age" is a domain index $K$; patients of different ages belong to different domains. One robust way to predict $Y$ is to use breathing patterns such as periodicity (periodic breathing signals usually indicate "Deep Sleep"); these patterns are encoded as **causal representations** $V$ in our CADA. The unobserved **spurious features** $U$ in sleep studies may include 'respiratory rate' mentioned in Example 1. Breathing signals $X$ from older patients, i.e., larger $K$, may have a lower 'respiratory rate' due to weakened respiratory muscles; therefore it is **not a reliable feature** for predicting $Y$. A neural network can extract the compact representation $R$ given the raw breathing signal $X$ as input, but $R$ still inherits the spurious features $U$ from $X$; therefore directly using $R$ to predict $Y$ would not generalize across domains (different $K$), as in Example 1. To address this problem, we would like to remove the spurious features $U$ from $R$ according to $K$, thereby obtaining the causal factors/representations $V$. We can then train a generalizable (and causally transportable) model to predict sleep stages $Y$ using the causal representations $V$.*

## 3.2 Challenges in Continuously Transportable Domain Adaptation

**Conditional Model $P(Y|X)$ Is Not Generalizable across Domains.** Due to the confounding factors $K$ and $U$ between $X$ and $V$, the computable quantity $P(Y|X)$ trained in source domains usually do not generalize to target domains, i.e. $P(Y|X, K = k_1) \neq P(Y|X, K = k_2)$ for $k_1 \neq k_2 \in \mathcal{K}$.

**Causal Model $P(Y|do(X))$ Is Generalizable across Domains.** Fortunately, with the causal graph in Fig. 1, the causal effect $P(Y|do(X))$ is transportable (and generalizable) across different domains. This is demonstrated in Theorem 3.1 and Theorem 3.2 below (more details in Appendix A).

**Theorem 3.1** (**Transportability of $P(v|do(x))$**). *With the causal diagram in Fig. 1, the causal relation $P(v|do(x))$ is transportable from domain $k_1$ to $k_2$ for any $k_1 \neq k_2$.*

**Theorem 3.2** (**Transportability of $P(y|do(x))$**). *With the causal diagram in Fig. 1, the causal relation $P(y|do(x))$ is trivially transportable from domain $k_1$ to $k_2$ for any $k_1 \neq k_2$ using:*

$$P(y|do(x)) = \sum_v P(y|v)Q(v|x), \tag{1}$$

*where $Q(v|x)$ is an probabilistic encoder generating encoding $v$ given the input $x$. It is constructed as*

$$Q(v|x) = \sum_r P(r|x) \sum_{x'} P(v|r, x')P(x'). \tag{2}$$

**Transportability Alone Is Not Sufficient.** From Eqn. 1, we can see that $v$ computed from Eqn. 2 can be treated as the encoding for the input $x$. In practice, due to covariate shift (Ganin et al., 2016; Ben-David et al., 2010), there is no guarantee that directly using Eqn. 1 will lead to good domain adaptation performance. Specifically, Eqn. 2 may theoretically guarantee causal correlation learning, such guarantees break in the presence of covariate shift, which leads to non-positivity in variable $V$'s probability distributions and non-identifiability of the causal interventional distribution $P(V|do(X))$ (Definition 3.2.3 and Definition 3.2.4 of (Pearl et al., 2000)). The following lemma adapted from (Ben-David et al., 2010) shows the consequence of covariate shift.

**Lemma 3.1** (**Target-Domain Error Bound**). *Let $\mathcal{H}$ be a hypothesis space, and $h \in \mathcal{H} : \mathcal{V} \to \{0, 1\}$. $\mathcal{D}_S(V)$ and $\mathcal{D}_T(V)$ are the encoding distributions of the source and target domains, respectively. We have:*

$$\epsilon_{\mathcal{D}_T}(h) \leq \epsilon_{\mathcal{D}_S}(h) + \tfrac{1}{2}d_{\mathcal{H}\Delta\mathcal{H}}(\mathcal{D}_S(V), \mathcal{D}_T(V)) + \lambda,$$

*where $\epsilon_{\mathcal{D}_S}(h)$ and $\epsilon_{\mathcal{D}_T}(h)$ are the prediction error in the source and target domains, respectively. The constant $\lambda = \min_h(\epsilon_{\mathcal{D}_S}(h) + \epsilon_{\mathcal{D}_T}(h))$. The $\mathcal{H}\Delta\mathcal{H}$ divergence $d_{\mathcal{H}\Delta\mathcal{H}}(\mathcal{D}_S(V), \mathcal{D}_T(V))$ characterizes the divergence between the source domain's and the target domain's encoding distributions.*

Lemma 3.1 shows that the generalization error of a target domain is bounded by the source-domain error, the distribution divergence between the encoding $v$ of the source and target domains, and the constant $\lambda$. Therefore it is also important to reduce the second term $d_{\mathcal{H}\Delta\mathcal{H}}(\mathcal{D}_S(V), \mathcal{D}_T(V))$ by aligning the encoding distributions of $v$ from different domains.

**Definition 1** (**Ideal Encoder $Q(v|x)$**). *With the analysis above, an ideal encoder $Q(v|x)$ that encodes the input $x$ into an encoding $v$ is an encoder that*

*(1) has the form of Eqn. 2 and*
*(2) ensures the marginal independence $k \perp\!\!\!\perp v$.*

The first requirement in Definition 1 is trivially satisfied by construction. However, with $v$ generated by $Q(v|x)$ in Eqn. 2, there is no guarantee that $k \perp\!\!\!\perp v$ (the second requirement in Definition 1); this is shown in Theorem 3.3 below.

**Theorem 3.3.** *With the causal diagram $G$ in Fig. 1, $K$ is not independent with $V$.*

Therefore, to enable transportable domain adaptation, one needs to train an encoder parameterized by Eqn. 2 while aligning the distributions of $v$ for different domains. Such alignment is done via an adversarial training process, as detailed in Sec. 4 below.

## 4 METHOD

To learn an ideal encoder $Q(v|x)$ that satisfies the two requirements as defined in Definition 1, we propose to learn an sophisticated encoder $Q(v|x)$ in the form of Eqn. 2 while making sure that the distribution of the causal encodings $v \sim Q(v|x)$ from all domains $\mathcal{K}$ are aligned. Such alignment corresponds to Requirement (2) of Definition 1, i.e., the marginal independence $k \perp\!\!\!\perp v$; it ensures that all labels can be accurately predicted by the shared predictor $P(y|v)$ using Eqn. 1. We achieve this using an additional discrimininator $D(k|v)$, which predicts the domain index $k$ given the causal encoding $v$.

Once an ideal encoder $Q(v|x)$ is trained, one can then use Eqn. 1 to predict $y$. Below, we starts by introducing our CADA's causal encoder $Q(v|x)$, predictor $F(y|v)$, and discriminator $D(k|v)$. We can put them together to form a final objective function to perform a minimax optimization.

**Causal Encoder $Q(v|x)$.** Our causal encoder will take the form of Eqn. 2, which consists of three components, i.e., an intermediate encoder $P(r|x)$, an augmented encoder $P(v|r, x)$, and a data sampler $P(x')$:

- **Intermediate Encoder** $P(r|x)$**:** The intermediate encoder encodes the input $x$ into the intermediate representation $r$. To enable sampling of $r$, this is a *probabilistic* encoder, where sampling $r \sim P(r|x)$ equivalent to

$$r \sim \mathcal{N}\big(\mu_r(x), \sigma_r^2(x)\big),$$

  where $\mu_r(\cdot)$ and $\sigma_r^2(\cdot)$ denote two neural networks taking $x$ as input and predict the mean and variance of $r$, respectively. These networks can be trained using the reparameterization trick (Kingma & Welling, 2013).
- **Augmented Encoder** $P(v|r, x')$**:** The augmented encoder takes as input the intermediate representation $r$ and another input data point $x' \neq x$ as augmented data to predict the preliminary $v$. Similar to $P(r|x)$, sampling $v \sim P(v|r, x')$ equivalent to

$$v \sim \mathcal{N}\big(\mu_v(r, x'), \sigma_v^2(r, x')\big),$$

  where $\mu_v(\cdot, \cdot)$ and $\sigma_v^2(\cdot, \cdot)$ denote two neural networks trained using the reparameterization trick (Kingma & Welling, 2013).
- **Data Sampler** $P(x')$**:** The data sampling uniformly randomly samples data from different domains with different labels. The goal is to include diverse data with different spurious features into the model, such that these spurious features can cancel each other out using the summation $\sum_{x'} P(v|r, x')P(x')$ in Eqn. 2, thereby leading to "more causal" encodings $v$.

As a result, given the input $x$, sampling $v$ from our causal encoder is equivalent to first sampling $x$'s intermediate representation $r$ from $P(r|x)$, sampling $x'$ from the data sampler $P(x')$, and then sampling $v$ from $P(v|r, x')$. Note that this composed causal encoder $Q(v|x)$ can be trained end-to-end with the reparameterization trick (Kingma & Welling, 2013).

**Predictor** $F(y|v)$**.** According to Eqn. 1, our predictor $F(y|v)$ takes as input the causal encoding $v$ from $Q(v|x)$ and predicts the label $y$. In CADA, $F(y|v)$ is parameterized by a simple multi-layer perceptron (MLP). Note that $F(y|v) = P(y|v)$ in CADA.

**Discriminator** $D(k|v)$**.** Our discriminator $D(k|v)$ takes as input the causal encoding $v$ and predicts the domain index $k$. $D(k|v)$ is crucial in terms of aligning the distributions of causal encodings $v$ from different domains (more details below). Since we focus on DA across continuously indexed domains, $D(k|v)$ will directly *regress* the continuous index $k$ (Wang et al., 2020) rather than performing classification (Ganin et al., 2016).

**Final Objective Function.** Putting $Q(v|x)$, $F(y|v)$, and $D(k|v)$ together, we can form the following minimax optimization:

$$\min_{Q,F} \max_D V_p(Q, F) - \lambda_d V_d(D, Q), \tag{3}$$

where the instantiations of both terms are **very different** from (Ganin et al., 2016; Wang et al., 2020). Specifically, the **first term**

$$V_p(Q, F) \triangleq \mathbb{E}^s[L_p(\widehat{y}, y)], \tag{4}$$

$$V_p(Q, F) \triangleq \mathbb{E}^s[L_p(\widehat{y}, y)] \approx \frac{1}{N_s} \sum_{(x,y,k)} -\log \sum_v F(y|v)Q(v|x), \tag{5}$$

measures the *difference* between the *prediction* $\widehat{y}$ (computed by $Q$ and $F$) and the *ground truth* $y$; this can be computed as the negative log-likelihood of $y$, i.e., $-\log \sum_v F(y|v)Q(v|x)$. We compute the average difference over $N_s$ tuples of $(x, y, k)$ sampled from $p^s(x, y, k)$.

Note that to generate the prediction $\widehat{y}$, one has to go through $Q(v|x)$ and $F(y|v)$, with $Q(v|x)$ consisting of three components $P(r|x)$, $P(x')$, and $P(v|r, x')$ in Eqn. 2. Specifically, given an input-label-index tuple $(x, y, k)$ sampled from the source data distribution, we will first sample $r$ from $P(r|x)$, sample $x'$ from $P(x')$, sample $v$ from $P(v|r, x')$ given the previous sampled $r$ and $x'$, and then generate prediction $\widehat{y}$ from $P(y|v)$ given the sampled $v$. We then compute the prediction loss for $\widehat{y}$ against the label $y$.

The **second term**,

$$V_d(D, Q) \triangleq \mathbb{E}[L_d(\widehat{k}, k)],$$

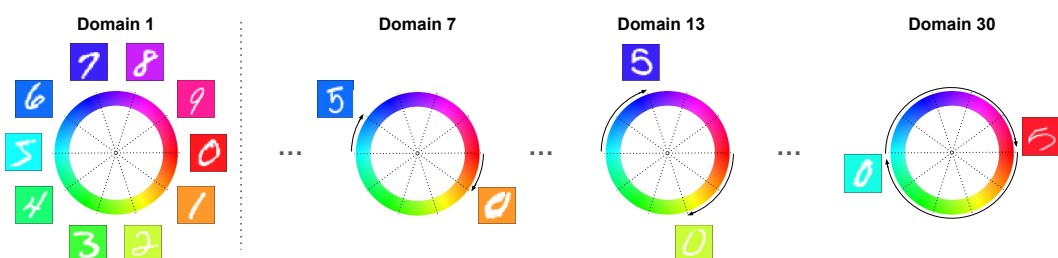

Figure 2: Illustration of the C2MNIST dataset. As shown on the left, the background colors for digit 0 to 9 are evenly separated on the color wheel. As we rotate $6(k-1)°$ clockwise from the base angles (domain 1), the background colors for domain $k$ shift while still maintaining even separation.

$$V_d(D,Q) \triangleq \mathbb{E}[L_d(\widehat{k}, k)] \approx \frac{1}{N_{all}} \sum_{(x,y,k)} - \log \sum_v D(k|v)Q(v|x), \tag{6}$$

measures the *difference* between the *predicted* domain index $\widehat{k}$ (computed by $Q$ and $D$) and the *ground-truth* domain index $k$; this can be computed as the negative log-likelihood of $k$, i.e., $-\log \sum_v D(k|v)Q(v|x)$. Similar to Eqn. 5, we compute the average difference over $N$ data points sampled from $p(x, y, k)$. Specifically, given an input-label-index tuple $(x, y, k)$ sampled from the entire data distribution, we first sample $v$ from $Q(v|x)$ by going through $P(r|x)$, $P(x')$, and $P(v|r, x')$; the discriminator $D(k|v)$ then generates the predicted domain index $\widehat{k}$ given $v$.

**Two Requirements in Definition 1.** The minimax optimization with the objective function Eqn. 3 ensures that two requirements in Definition 1 are satisfied:

- Requirement (1) of Definition 1 is ensured by parameterizing the causal encoder $Q(v|x)$ according to Eqn. 2, which uses two probabilistic neural networks, i.e., the intermediate encoder $P(r|x)$ and the augmented encoder $P(v|r, x')$, and the data sampler $P(x')$.
- Requirement (2) of Definition 1 is encouraged by the adversarial loss, i.e., the minimax optimization; the discriminator $D(k|v)$ will be trained to predict the domain index $k$, while causal encoder $Q(v|x)$ is trained to fool the discriminator. This adversarial process will try to align the distributions of $v$ across different domains such that $D(k|v)$ cannot accurately predict the domain index $k$. As shown in (Wang et al., 2020), the solution where $k \perp\!\!\!\perp v$ will be among the equilibria of the minimax optimization in Eqn. 3.

**Inference (Prediction).** After our CADA is trained using Eqn. 3, one can then make predictions (causal inference) by combining Eqn. 1 and Eqn. 2. Specifically, given the input $x$, one can predict $y$ using

$$P(y|do(x)) = \sum_v P(y|v) \sum_r P(r|x) \sum_{x'} P(v|r, x')P(x'),$$

where we use Monte Carlo estimation to draw samples from different conditional distributions and aggregate the results to obtained the final prediction (see algorithm 1 in Appendix B for the detailed algorithm).

## 5 EXPERIMENTS

We evaluated our method, CADA, using a semi-synthetic image dataset, Continuous Colored-MNIST (C2MNIST), along with two real-world medical datasets, Sleep Heart Health Study (SHHS) (Quan et al., 1998) and Multi-Ethnic Study of Atherosclerosis (MESA) (Chen et al., 2014), where continuously shifting spurious features are introduced. These empirical studies corroborate the theoretical discoveries outlined earlier and demonstrate the following:

- Using categorical domain adaptation to align continuously indexed domains with continuously shifting spurious features leads to suboptimal alignment and performance.
- Continuously indexed domain adaptation methods alone tend to align shifting spurious features rather than causal features, and are therefore not effective for domain adaptation with continuously shifting spurious features.
- Causality-based domain adaptation methods alone suffer from covariant shift across domains and are not effective in adaptation across continuously indexed domains.

Table 2: **C2MNIST accuracy (%) for CADA and various baselines.** We report the accuracy in the source domains and each target domain range. The intervals in the first row represent the domain range of corresponding domains. The average accuracy across target domains is shown in the last column. We use **bold face** to highlight the best results.

| Method | $[1, 7)$ (Source) | $[7, 11)$ | $[11, 15)$ | $[15, 19)$ | $[19, 23)$ | $[23, 27)$ | $[27, 31)$ | Average |
|---|---|---|---|---|---|---|---|---|
| Source-Only | **100.0** | 0.1 | 0.0 | 0.0 | 0.0 | 0.0 | 0.0 | 0.0 |
| CUA | 97.3 | 13.25 | 0.0 | 0.0 | 0.0 | 0.0 | 0.0 | 2.2 |
| ADDA | **100.0** | 1.4 | 0.0 | 0.0 | 0.0 | 0.0 | 0.0 | 0.2 |
| DANN | **100.0** | 46.0 | 5.0 | 0.0 | 0.0 | 0.0 | 0.0 | 8.5 |
| CDANN | 99.9 | 76.1 | 62.0 | 24.9 | 9.8 | 11.5 | 7.8 | 32.0 |
| MDD | **100.0** | 66.1 | 35.5 | 20.0 | 9.3 | 0.5 | 5.4 | 22.8 |
| CIDA | **100.0** | 1.8 | 0.0 | 0.0 | 0.0 | 0.0 | 0.0 | 0.3 |
| VOOD | **100.0** | 16.8 | 9.6 | 8.9 | 9.0 | 10.0 | 8.8 | 10.6 |
| GDA | 74.5 | 3.9 | 3.6 | 2.5 | 4.4 | 4.5 | 2.6 | 3.6 |
| CADA (Ours) | **100.0** | **100.0** | **100.0** | **100.0** | **100.0** | **100.0** | 82.9 | **97.1** |

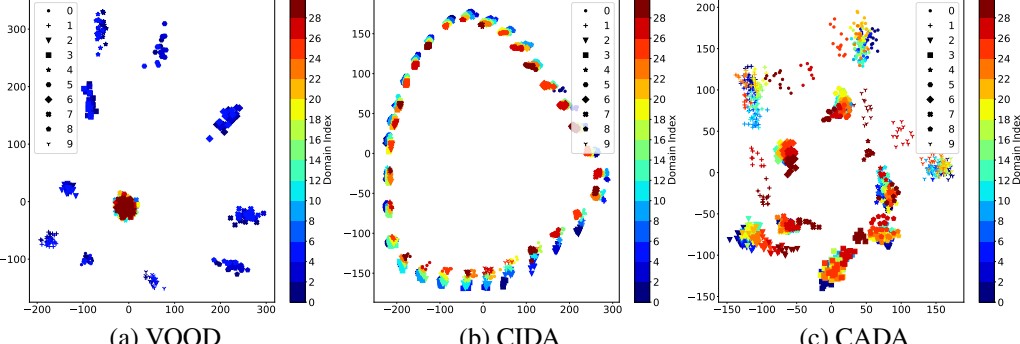

| (a) VOOD | (b) CIDA | (c) CADA |
|---|---|---|

Figure 3: Visualization of learned representations on the C2MNIST dataset, with dimensionality reduction to 2D using principle component analysis (PCA). Domain indices are indicated by color, while data labels are represented by various markers. **(a)** VOOD's source- and target-domain embedding distributions are not aligned due to covariate shift, resulting in poor performance in the target domain. **(b)** CIDA's source- and target-domain embedding distributions are better aligned compared to VOOD. However, they are not aligned by labels. Specifically, these embeddings form 30 clusters, with each cluster containing embeddings with different labels, resulting in poor classification performance. **(c)** CADA's source- and target-domain embedding distributions are aligned by labels. Specifically, these embeddings form 10 clusters, with each cluster containing embeddings with the same label; this makes learning an optimal decision bounary for classification significantly easier, resulting in superior classification performance. See enlarged versions of these figures in Appendix I.

- Our CADA successfully infer and align causal representations from continuously indexed domains even in the presence of continuously shifting spurious features, thereby significantly improving performance.

## 5.1 BASELINES AND IMPLEMENTATION DETAILS

We compare CADA with state-of-the-art methods in domain adaptation and causal transportation baselines, including Continuous Unsupervised Adaptation (**CUA**) (Bobu et al., 2018), Adversarial Discriminative Domain Adaptation (**ADDA**) (Tzeng et al., 2017), Domain Adversarial Neural Network (**DANN**) (Ganin et al., 2016), Conditional Domain Adversarial Neural Network (**CDANN**) (Zhao et al., 2017), Margin Disparity Discrepancy (**MDD**) (Zhang et al., 2019), Continuously Indexed Domain Adaptation (**CIDA**) (Wang et al., 2020), Visual Out-of-Distribution Generalization (**VOOD**) (Mao et al., 2022), and Gradual Domain Adaptation (**GDA**) (He et al., 2024) Please refer to Appendix E for more implementation details.

## 5.2 CONTINUOUS COLORED-MNIST (C2MNIST)

**Dataset Description.** We start from the simple MNIST (LeCun et al., 1998) dataset since the simplicity of the dataset allows us to study the methods in a controlled environment. We adapt

Table 3: **Accuracy (%) for different methods in the SHHS dataset.** The task is to transfer sleep stage prediction models from patients in the age range $[44, 53)$ to $[53, 90]$. We mark the best result with **bold face** and the second best results with underline.

| Method | [44, 53) (Source) | [53, 58) | [58, 63) | [63, 68) | [68, 73) | [73, 78) | [78, 83) | [83, 90) | Average |
|---|---|---|---|---|---|---|---|---|---|
| Source-Only | 88.4 | 62.5 | 61.8 | 59.2 | 59.1 | 59.7 | 59.8 | 59.9 | 60.2 |
| CUA | 85.7 | 73.6 | 74.9 | 73.0 | 72.7 | 71.1 | 67.8 | **67.2** | 71.1 |
| ADDA | 85.6 | 73.9 | 74.1 | 71.9 | 71.6 | 69.9 | 66.6 | 65.4 | 70.1 |
| DANN | 87.0 | 72.4 | 72.5 | 69.1 | 67.5 | 64.6 | 60.9 | 59.9 | 66.2 |
| CDANN | 86.3 | 72.6 | 72.8 | 69.6 | 68.1 | 65.5 | 61.6 | 60.3 | 66.7 |
| MDD | **91.7** | 69.8 | 69.8 | 70.1 | 71.2 | 69.8 | 67.4 | 66.6 | 69.0 |
| CIDA | 86.0 | 72.7 | 72.4 | 69.4 | 69.5 | 68.6 | 66.5 | 65.4 | 68.9 |
| VOOD | 86.7 | 60.1 | 61.6 | 62.5 | 63.6 | 63.6 | 63.3 | 59.0 | 61.7 |
| GDA | 84.2 | 75.3 | 77.0 | 73.9 | 72.1 | 68.5 | 63.8 | 61.3 | 69.6 |
| CADA (Ours) | 84.3 | **76.7** | **77.7** | **76.1** | **74.7** | **72.1** | **68.9** | 67.1 | **72.8** |

Table 4: **Accuracy (%) for different methods in the MESA dataset.** The task is to transfer sleep stage prediction models from patients in the age range $[54, 59)$ to $[89, 92]$. We mark the best result with **bold face** and the second best results with underline.

| Method | [54, 59) (Source) | [59, 64) | [64, 69) | [69, 74) | [74, 79) | [79, 84) | [84, 89) | [89, 92] | Average |
|---|---|---|---|---|---|---|---|---|---|
| Source-Only | 93.3 | 35.6 | 35.2 | 35.3 | 41.7 | 55.4 | 66.6 | 54.1 | 46.0 |
| CUA | 79.9 | **75.1** | **74.1** | 71.4 | 70.8 | 70.1 | 68.4 | 65.3 | 70.9 |
| ADDA | 88.7 | 65.3 | 70.7 | 70.7 | 72.1 | 72.6 | 68.3 | 61.3 | 68.9 |
| DANN | 89.3 | 59.6 | 58.6 | 60.0 | 63.7 | 66.8 | 66.9 | 66.6 | 63.1 |
| CDANN | 88.5 | 61.1 | 60.0 | 59.4 | 61.4 | 64.2 | 64.6 | 64.9 | 62.2 |
| MDD | 88.5 | 60.7 | 60.2 | 62.7 | 67.3 | 68.5 | 67.6 | 67.9 | 64.9 |
| CIDA | 87.8 | 61.7 | 60.8 | 61.2 | 64.7 | 66.5 | 66.0 | 64.7 | 63.6 |
| VOOD | **94.9** | 28.7 | 31.2 | 34.4 | 41.6 | 57.7 | 59.8 | 46.1 | 42.7 |
| GDA | 82.0 | 64.9 | 64.1 | 62.1 | 61.1 | 62.4 | 59.2 | 59.5 | 62.0 |
| CADA (Ours) | 80.8 | 74.3 | 74.0 | **72.8** | **73.2** | **72.9** | **72.3** | **71.1** | **73.0** |

the dataset to continuously indexed domain adaptation with shifting spurious features by adding background colors to the digits, where the color depends on both the domain index and the digit (more details in Appendix E). This leads to our new Continuous Colored-MNIST (C2MNIST) dataset. Fig. 2 shows some example images from different domains of C2MNIST.

**Accuracy.** We further divide the target domains into 6 parts for evaluation based on the distance from the source domains. Table 2 shows the results for different methods. It is evident that in target domains, CUA, ADDA, DANN, and CIDA are completely misled by the continuously shifting spurious features and therefore fail to make predictions in the distant domains. CDANN and MDD, while not entirely wrong in the distant domains, still perform no better than random guessing. While VOOD may theoretically guarantee causal correlation learning, such guarantee break in the presence of covariate shift, which leads to non-positivity in $V$'s probability distributions and non-identifiability of the causal interventional distribution $P(V|do(X))$ (Definition 3.2.3 and Definition 3.2.4 of (Pearl et al., 2000)); see Sec. 3.2 for detailed discussion. As a result, VOOD improves upon random guess only in target domains that are the closest to source domains, i.e., domains $[7, 11)$.

**Visualization of Latent-Space Representations.** To gain more insights on how CADA (1) aligns representation from different domains to remove covariate shift and (2) eliminates continuously shifting spurious features, we visualize the learned representation for two representative baselines: VOOD and CIDA, as well as our model CADA in Fig. 3. The domain indices are indicated by color, while labels are indicated by various markers.

- **VOOD: Failing to Remove Covariate Shift.** Fig. 3(a) shows VOOD's learned representations, which solely focus on causal correlation learning, unable to align the distributions of source domains (mostly blue points in the figure) and target domains (mostly non-blue points in the figure) due to covariate shift, resulting in poor performance in the target domain (detailed analysis in Sec. 3.2).
- **CIDA: Failing to Remove Continuously Shifting Spurious Features.** Fig. 3(b) shows the representations from CIDA, a typical continuously indexed domain adaptation method. CIDA aligns source- and target-domain embedding distributions better compared to VOOD. However,

Table 5: Ablation study on the C2MNIST dataset. The best results are highlighted in **bold**.

| Method | $[1, 7)$ (Source) | $[7, 11)$ | $[11, 15)$ | $[15, 19)$ | $[19, 23)$ | $[23, 27)$ | $[27, 31)$ | Average |
|---|---|---|---|---|---|---|---|---|
| Source-Only | **100.0** | 0.1 | 0.0 | 0.0 | 0.0 | 0.0 | 0.0 | 0.0 |
| CADA w/o Causal Encoder | **100.0** | 1.8 | 0.0 | 0.0 | 0.0 | 0.0 | 0.0 | 0.3 |
| CADA w/o Discriminator | **100.0** | 48.0 | 9.9 | 9.9 | 9.1 | 9.6 | 9.4 | 16.0 |
| CADA w/o Causal Inference | **100.0** | 3.6 | 0.0 | 0.0 | 0.0 | 0.0 | 0.0 | 0.6 |
| CADA (Full) | **100.0** | **100.0** | **100.0** | **100.0** | **100.0** | **100.0** | **82.9** | **97.1** |

they are not aligned by labels. Specifically, these embeddings form 30 clusters, with each cluster containing embeddings with different labels, resulting in poor classification performance.

- **CADA: Successfully Removing Both Covariant Shift and Continuously Shifting Spurious Features.** Fig. 3(c) shows the representations from our CADA. CADA successfully align source- and target-domain embedding distributions by labels. Specifically, these embeddings form 10 clusters, with each cluster containing embeddings with the same label; this makes learning an optimal decision bounary for classification significantly easier, resulting in superior classification performance. For better view, please see the enlarged versions of (a), (b), and (c) in Appendix I.

## 5.3 HEALTHCARE DATASETS

**Dataset Description.** We use two medical datasets, SHHS and MESA to evaluate different methods. Both datasets contain records of subjects' full-night breathing signals and corresponding sleep stage labels for every 30-second segment. Sleep stage labels include "Awake", "Light Sleep N1", "Light Sleep N2", "Deep Sleep", and "Rapid Eye Movement (REM)". Given a breathing signal segment $\mathbf{x}$, a common task in sleep studies is to predict the sleep stage label $y$. While prior studies may consider 'Light Sleep N1' and 'Light Sleep N2' to be a single stage (Zhao et al., 2017; Wang et al., 2020), we maintain a 5-class classification task in this paper. Among all the subjects' information, age can serve as a natural domain index. The age range of subjects used is $[44, 90]$ and $[54, 92]$ for SHHS and MESA, respectively. Our SHHS and MESA datasets contain continuously shifting spurious noise to simulate the potential increase in noise due to age-related lower-quality sleeping, breathing disorder (e.g., sleep apnea), etc.

**Accuracy.** Table 3 and Table 4 show the accuracy for different methods on SHHS and MESA, respectively. The target domains of both datasets are divided into 7 parts for evaluation based on the distance from the source domains. One observation is that VOOD, as a causal tranporation method, shows negative improvement compared to Source-Only (e.g., training the model on source domains and directly use it on target domains without adaptation). While methods that directly use categorical domain adaptation perform poorly in the normal continuously indexed domain adaptation setting, CUA shows better performance than other baselines in our setting. On the other hand, our CADA avoids the influence of continuously shifting spurious correlation, demonstrating stable and good performance in both datasets.

## 5.4 ABLATION STUDIES

We also conduct ablation studies at both the architectural and algorithmic levels. The results are summarized in Table 5. Removing the causal encoder, eliminating the discriminator, or disabling the causal inference mechanism all lead to substantial drops in accuracy, thereby verifying that causal representation learning, adversarial domain alignment, and explicit causal reasoning are each essential for robust generalization (more details in Appendix F).

## 6 CONCLUSION

In this paper, we identify the problem of continuous spurious shift and propose continuously transportable domain adaptation (CADA) as the first general DA method to address this problem. Our theoretical analysis shows that our CADA can better ensure causal transportability across continuously indexed domains. Empirical results on both semi-synthetic and real-world medical datasets show that our method outperforms the state-of-the-art DA methods in the face of continuous spurious shift. Interesting future work includes extending the proposed method to multi-dimensional domain indices, more complex shifting spurious features, and other applications beyond healthcare.

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

## A  PROOFS

**Theorem A.1 (Transportability of $P(v|do(x))$).** *With the causal diagram in Fig. 1, the causal relation $P(v|do(x))$ is transportable from domain $k_1$ to $k_2$ for any $k_1 \neq k_2$.*

*Proof.* Given that $R$ satisfies the front-door criterion (Pearl, 2009), we can then use the front-door adjustment formula to obtain:

$$P(v|do(x)) = \sum_r P(r|x) \sum_{x'} P(v|r, x')P(x').$$

By the definition of trivial transportability (Pearl & Bareinboim, 2011), since $P(v|do(x))$ is identifiable, we have that $P(v|do(x))$ is transportable. □

**Theorem A.2 (Transportability of $P(y|do(x))$).** *With the causal diagram in Fig. 1, the causal relation $P(y|do(x))$ is trivially transportable from domain $k_1$ to $k_2$ for any $k_1 \neq k_2$ using:*

$$P(y|do(x)) = \sum_v P(y|v)Q(v|x), \tag{7}$$

*where $Q(v|x)$ is an probabilistic encoder generating encoding $v$ given the input $x$. It is constructed as*

$$Q(v|x) = \sum_r P(r|x) \sum_{x'} P(v|r, x')P(x'). \tag{8}$$

*Proof.* Using the chain rule we have that $P(y|do(x)) = \sum_v P(y|v)P(v|do(x))$, where $P(v|do(x))$ is given by Theorem A.1. Denoting $P(v|do(x))$ as $Q(v|x)$ concludes the proof. □

**Theorem A.3.** *With the causal diagram $G$ in Fig. 1, $K$ is not independent with $V$.*

*Proof.* This is straightforward given that $K \not\perp\!\!\!\perp V$ in $G_{\overline{X}}$. □

---

**Algorithm 1** Inference (Prediction) Using CADA

---

1: **Input:** Query $x$, data distribution $\mathcal{D}$ over $\{(x, y, k)\}$, causal encoder $Q(v|x)$ consisting of the intermediate encoder $P(r|x)$ and the augmented encoder $P(v|r, x')$. Numbers of samples for $r$, $x'$, and $v$, i.e., $N_r$, $N_x$, and $N_v$.
2: **for** $i = 1, ..., N_r$ **do**
3:    Sample $r^{(i)} \sim P(r|x)$.
4:   **for** $j = 1, ..., N_x$ **do**
5:      Sample from the data distribution: $\mathbf{x}'^{(ij)} \sim P(x')$.
6:     **for** $m = 1, ..., N_v$ **do**
7:       Sample $v^{(ijm)} \sim P(v|r^{(i)}, \mathbf{x}'^{(ij)})$.
8:     **end for**
9:   **end for**
10: **end for**
11: Compute the causal effect for each class: $\quad P(y|do(x)) \quad = \quad \frac{1}{N_r N_x N_v} \sum_{i=1}^{N_r} \sum_{j=1}^{N_x} \sum_{m=1}^{N_v} P(y|v^{(ijm)})$.
12: **Output:** Class prediction $\widehat{y} = \mathrm{argmax}_y P(y|do(x))$.

---

Table 6: Accuracy (%) and Time Cost (Milliseconds per Image) for Different Numbers of Samples.

| Model | # Samples | [1, 7) (Source) | [7, 11) | [11, 15) | [15, 19) | [19, 23) | [23, 27) | [27, 31) | Average | Time |
|---|---|---|---|---|---|---|---|---|---|---|
| VOOD (Baseline) | - | 100.0 | 48.0 | 9.9 | 9.9 | 9.1 | 9.6 | 9.4 | 16.0 | 0.40 ms |
| CIDA (Baseline) | - | 100.0 | 1.8 | 0.0 | 0.0 | 0.0 | 0.0 | 0.0 | 0.3 | 0.13 ms |
| CADA | 1 | 100.0 | 100.0 | 100.0 | 100.0 | 100.0 | 99.9 | 74.7 | 95.8 | 0.29 ms |
| CADA | 4 | 100.0 | 100.0 | 100.0 | 100.0 | 100.0 | 100.0 | 76.5 | 96.1 | 0.31 ms |
| CADA | 9 | 100.0 | 100.0 | 100.0 | 100.0 | 100.0 | 100.0 | 77.2 | 96.2 | 0.35 ms |
| CADA (Original) | 55 | 100.0 | 100.0 | 100.0 | 100.0 | 100.0 | 100.0 | 82.9 | 97.1 | 0.41 ms |

## B   INFERENCE ALGORITHM OF CADA

Algorithm 1 shows details for CADA's inference (prediction) process.

## C   DEFINITIONS

Below we provide the formal definitions of spurious association and spurious shift in the context of CADA.

**Definition 2** (Spurious Association). *Two variables X and Y are spuriously associated if they are dependent in some context and there exist two other variables ($Z_1$ and $Z_2$) and two contexts ($S_1$ and $S_2$) such that:*

1. *$Z_1$ and X are dependent given $S_1$ (i.e. $Z_1 \not\perp\!\!\!\perp X | S_1$)*

2. *$Z_1$ and Y are independent given $S_1$ (i.e. $Z_1 \perp\!\!\!\perp Y | S_1$)*

3. *$Z_2$ and Y are dependent given $S_2$ (i.e. $Z_2 \not\perp\!\!\!\perp Y | S_2$)*

4. *$Z_2$ and X are independent given $S_2$ (i.e. $Z_2 \perp\!\!\!\perp X | S_2$)*

**Definition 3** (Spurious Shift). *Given a latent representation $r = (r_c, r_s)$, where $r_c$ captures the causal factors for predicting $y$, and $r_s$ captures spurious features, we say a spurious shift occurs when:*

$$P_{source}(y|r_c) = P_{target}(y|r_c), P_{source}(y|r_s) \neq P_{target}(y|r_s)$$

*That is, the causal mechanism is invariant, but the spurious correlations shift across domains.*

## D   COMPUTATIONAL COST

Table 6 shows the time cost for different models.

## E    MORE IMPLEMENTATION DETAILS

**Baselines.** Since the majority of the baselines are not designed for continuously indexed domains, we made slight generalizations to accommodate these baselines.

Specifically, for ADDA, MDD and VOOD, data with different domain indices are merged into one source and one target domain; for DANN, CDANN and CUA, the continuous domain spectrum is discretized into multiple disjoint domains, enabling adaptation between multiple source and target domains. In the case of CUA, the model adapts from the source domains to each target domain individually, progressing from the closest target to the farthest one

**C2MNIST Dataset.** We first create a color wheel from the HSV color space as shown in Fig. 2. For $k = 1$, colors are sampled from $0°, 36°, \dots, 324°$ of the color wheel for digit $0, 1, \dots, 9$ respectively. The angles are shown in dotted lines in the figure. For domain $k > 1$, the sampling angles of all the digits are rotated $6(k-1)°$ clockwise from the base domain $k = 1$. The difference in domain index therefore indicate the distance in the distribution of background colors.

We then treat the domains with $k \leq 6$ as the source domains, and use the remaining ones as the target domains. As a result, for any two distinct digits in the source domains, the ranges their background colors are disjoint, which means that background colors are highly predictive in the source domains. However, these background colors do not generalize to target domains; they are therefore spurious correlations that do not hold across domains. This allows us to verify the empirical performance of different domain adaptation methods.

**Compute Resources.** All methods are implemented with PyTorch (Paszke et al., 2019), and run on a single NVIDIA RTX A5000 GPU.

**Training Process.** For the C2MNIST dataset, all models are trained for 1600 epochs. Our CADA framework's training process is divided into four stages of 400 epochs each, transferring progressively from the previously learned domain to the next: $[7, 13)$, $[13, 19)$, $[19, 25)$ and $[25, 31)$. For healthcare datasets, all models are trained for 50 epochs. Our CADA framework's training process occurs in a single stage, transferring from the source domains to all target domains. We applied a simple grid search to determine the optimal configuration, which is fixed across methods for fair comparison.

**Model Architectures.** For fair comparison, we adopt the same backbone neural network architectures for baseline methods and CADA within the same dataset. In C2MNIST, we use the same multi-layer perceptron as in CIDA (Wang et al., 2020), with the hidden dimension set to 800. For healthcare datasets, we adopt the same setting as in CIDA for sleep learning, with the hidden dimension set to 384.

## F    ABLATION STUDY DETAILS

**Architectural Ablations: CADA w/o the Causal Encoder and CADA w/o the Discriminator**. In the first variant, i.e., CADA w/o the causal encoder we replace the causal encoder in CADA with a standard encoder to examine the role of causal representation learning. This ablation essentially corresponds to CIDA. In the second variant, i.e., CADA w/o the discriminator, we remove the discriminator from CADA to assess the contribution of adversarial domain alignment.

**Algorithmic Ablation: CADA w/o Causal Inference**. We also implemented a variant of CADA without causal inference; it uses the causal encoder but optimizes $P(y|x)$ instead of $P(y|do(x))$. This comparison isolates the explicit benefit of causal reasoning in mitigating spurious correlations.

**Results.** The results are summarized in Table 5. Removing the causal encoder, eliminating the discriminator, or disabling the causal inference mechanism all lead to substantial drops in accuracy. These results indicate that causal representation learning, adversarial domain alignment, and explicit causal reasoning are each essential for robust generalization.

Overall, the ablation experiments demonstrate that both the architectural components and the algorithmic causal inference mechanism provide complementary benefits, and their integration is critical for the robustness and effectiveness of CADA.

## G    LIMITATION

Our method currently only works with image and time series data. Additional work is needed to extend it to other modalities, such as text, videos, and audio.

## H    ACKNOWLEDGMENTS OF LLM USAGE

We acknowledge the use of ChatGPT 4o solely for language editing and polishing.

## I    ENLARGED FIGURES FOR REPRESENTATION VISUALIZATION

Fig. 4, Fig. 5, and Fig. 6 are the enlarged versions of Fig. 3(a), Fig. 3(b), and Fig. 3(c) in the main paper.

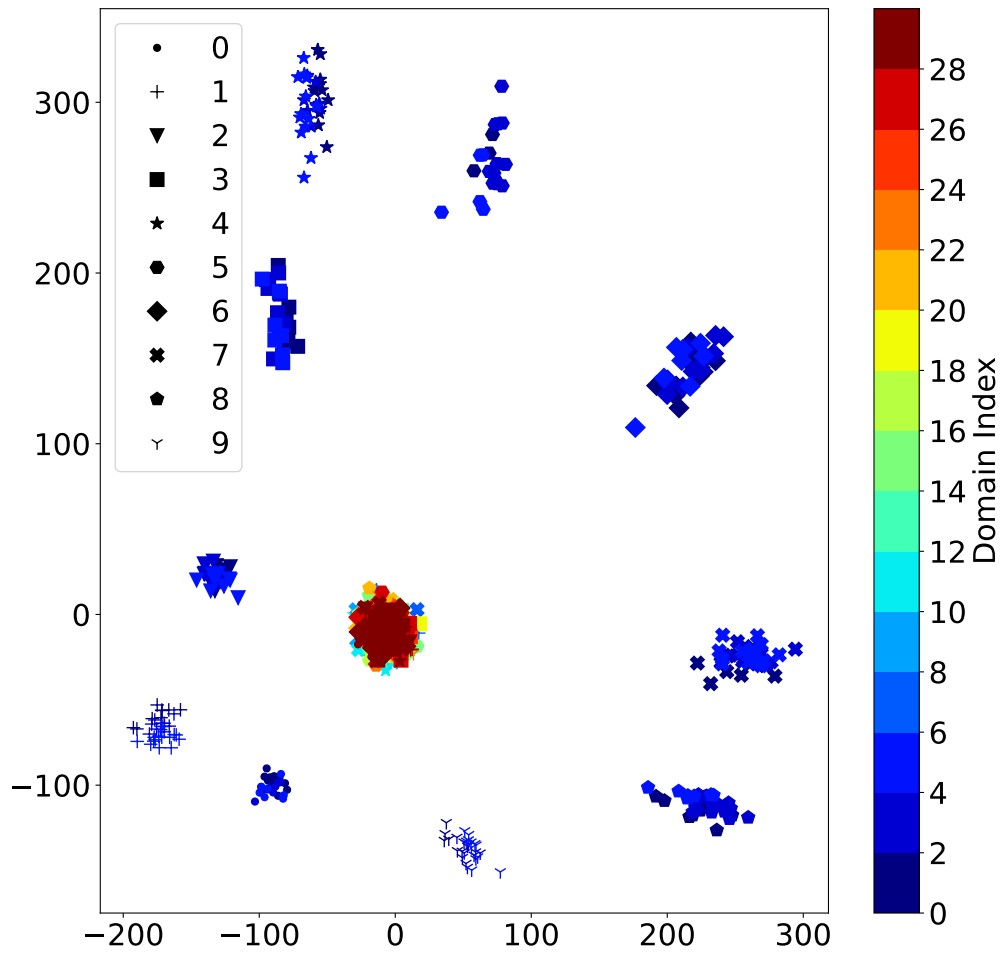

Figure 4: Enlarged version of Fig. 3(a): Visualization of VOOD's representations.

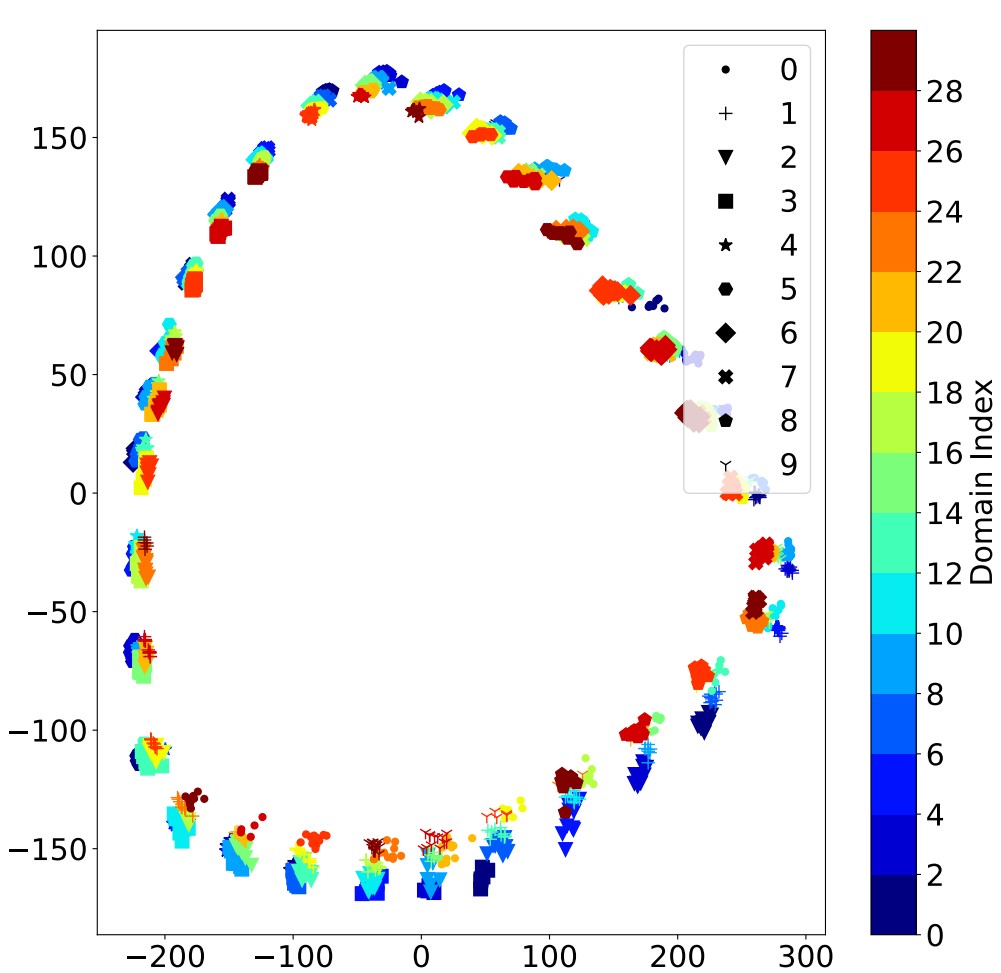

Figure 5: Enlarged version of Fig. 3(b): Visualization of CIDA's representations.

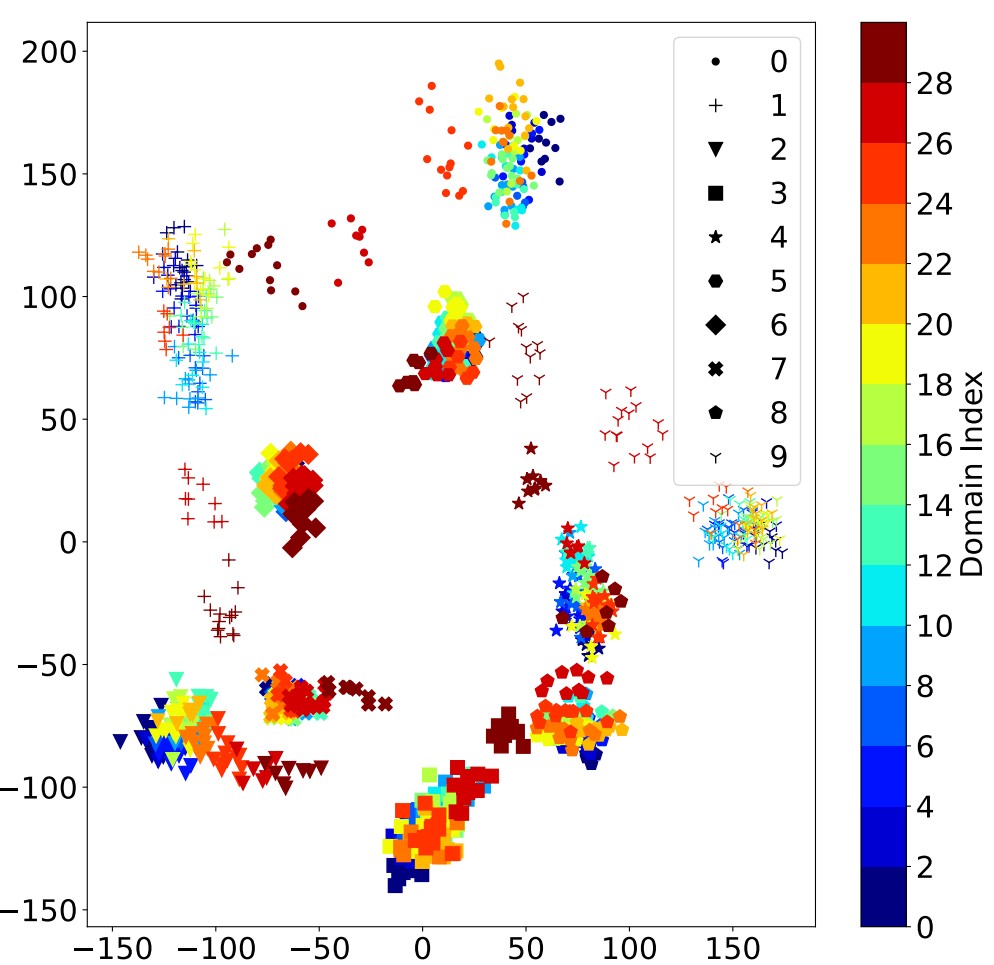

Figure 6: Enlarged version of Fig. 3(c): Visualization of CADA's representations.

