# OpenReview forum: "Toward Reliable Domain Adaptation under Continuous Spurious Shift"
_ICLR.cc/2026/Conference — ICLR 2026 Conference Withdrawn Submission_

### Official Review · Reviewer_Sjm5 · 2025-10-22

**Soundness:** 2
**Presentation:** 3
**Contribution:** 2
**Rating:** 2
**Confidence:** 4

**Summary:**

The paper studies continuously indexed domain adaptation (CIDA), arguing that continuously drifting spurious features can undermine existing methods. The authors cast the problem in a causal framework and primarily invoke the front-door criterion. They present theoretical results suggesting that the method proposed by authoors(CADA)  improves causal transportability across continuously indexed domains. Experiments on a semi-synthetic dataset and two real-world health datasets (SHHS and MESA) are good but not yet conclusive.

**Strengths:**

* The theory is correct; I have checked the proof.
* The result on the C2MNIST dataset is good.
* The writing is clear, and the causal model is reasonable.

**Weaknesses:**

* The theory shows limited innovation, as it mainly builds on existing theorems in the causal graph proposed in the paper.
* In the semi-synthetic image dataset, it seems to be just a dataset with shifting spurious features, which does not clearly relate to the causal model.
* For real-world datasets such as SHHS and MESA, in paper [1], other methods achieve better accuracy, while PCIDA shows state-of-the-art performance.

[1] Hao Wang, Hao He, and Dina Katabi. *Continuously Indexed Domain Adaptation.* In *ICML*, 2020.

**Questions:**

* (a) In the semi-synthetic image dataset, how does the construction demonstrate the causal model? To support the claim, some controlled synthetic experiments are needed.
* (b) Why are the baseline results in your paper significantly lower?
* (c) More experiments are needed to validate the performance on continuously indexed domain adaptation problems.

---

### Official Review · Reviewer_8xte · 2025-10-30

**Soundness:** 1
**Presentation:** 2
**Contribution:** 1
**Rating:** 2
**Confidence:** 4

**Summary:**

The paper introduces CADA (Continuously trAnsportable Domain Adaptation), a framework aimed at handling continuous spurious shifts in continuously indexed domain adaptation (CIDA) problems. It combines causal transportability analysis with adversarial domain alignment to learn “causally transportable” encodings that remain invariant across continuously changing domains. Theoretical discussions connect the method to front-door adjustment and transportability (Pearl, 2009), and experiments are conducted on semi-synthetic (C2MNIST) and two medical datasets (SHHS and MESA). While results show consistent performance improvements over baselines, the proposed formulation and experiments exhibit several weaknesses that undermine the novelty and rigor of the work.

**Strengths:**

-	Addresses a plausible and practically relevant problem (continuous spurious shifts).

-	Attempts to integrate causal reasoning with domain adaptation, a direction of general interest.

-	Provides complete experiments across both synthetic and medical datasets, with some consistency in results.

**Weaknesses:**

-	The method is a trivial extension of existing CIDA + DANN frameworks. The causal encoder and discriminator setup are directly borrowed from adversarial DA literature, and the “causal” aspect is only notational.

-	Theorems simply restate known transportability claims (Pearl, 2009) without proof of novel conditions or improvements. The reasoning around Eqn. (1–2) is tautological.

-	On real datasets (SHHS, MESA), performance differences are marginal (≈1–2%) and lack statistical testing; 100% accuracy on synthetic C2MNIST indicates overfitting or unrealistic setup.

-	Despite claims, there is no experiment demonstrating invariant causal effect recovery or robustness to interventions.

-	Key recent causal DG works (IRM, VREx, CORAL++) are missing.

-	The paper repeatedly uses “first method” or “theoretical guarantee” without substantiating either.

-	Many hyperparameters, dataset preprocessing, and implementation details are deferred to appendices, with no code or seed disclosure.

**Questions:**

-	How does the causal encoder differ, mathematically or empirically, from simply adding noise augmentation across domain indices as in standard adversarial alignment?

-	Can you formally show that Eqn. (2) leads to identifiable causal features under continuous shift?

-	Why does CADA achieve exactly 100% accuracy on all C2MNIST ranges—was there label leakage or an overly simplified setup?

-	Could results generalize to high-dimensional real-world settings beyond sleep datasets?

---

### Official Review · Reviewer_81SG · 2025-10-31

**Soundness:** 3
**Presentation:** 1
**Contribution:** 2
**Rating:** 4
**Confidence:** 3

**Summary:**

The authors propose a new domain adaptation algorithm based on a minimax objective that aims to find a new "causal factors" representation of an input by minimizing prediction error on the main task while maximizing prediction error on a domain prediction task. The form of generator for the "causal factors" representation is derived (including theoretical proofs) from a model where an input generates a base representation and the "causal factors" representation is derived from that given knowledge of other inputs.

**Strengths:**

* Outperforms by a large margin a variety of existing domain adaptation techniques on a synthetic dataset that injects a confounding color variable into MNIST
* Outperforms by a small margin a variety of existing domain adaptation techniques on two sleep stage prediction datasets, SHHS and MESA, where confounding noise has been injected.
* Includes an ablation that shows that all components of the approach are important for solving the color-confounded MNIST problem.

**Weaknesses:**

* The article is very difficult to read. Critical components of the theorems, e.g., do(x), are not defined, and the intuitions behind the theorems are not well explained. While I appreciate that the appendix contains formal proofs, the main text of the article needs to stand alone such that the proposed approach is understandable without the appendix. The article needs a high level explanation of the proposed algorithm and the intuitions for why it should take the form that it does before it begins diving into theorems and proofs.
* It doesn't seem like there's any evaluation on a real-world dataset. Though the article claims "Empirical results on... real-world medical datasets" it also says that "SHHS and MESA datasets contain continuously shifting spurious noise to simulate..." Thus it appears that the spurious noise in these datasets is simulated, not natural. Especially given how small the benefit of the approach is over CUA on the SHHS and MESA datasets, the reader is left wondering whether the approach is useful in the real world or only on datasets with synthetically generated noise.
* The proposed approach requires a Monte Carlo estimation with three nested sampling loops for each prediction, but computational cost is not discussed in the main text. There is a table in appendix D, but no description of the setup for that table's experiments, nor an explanation of why the proposed model, which requires 55 samples per prediction isn't slower than the other 2 models that don't appear to require any samples. The table is also critically missing the efficiency of CUA, the next best model according to Tables 3 and 4.
* Figures are often far away from where they are discussed, causing the reader to have to page back and forth over multiple pages.

**Questions:**

See above.

---

### Official Review · Reviewer_DniX · 2025-11-01

**Soundness:** 2
**Presentation:** 2
**Contribution:** 2
**Rating:** 2
**Confidence:** 5

**Summary:**

This paper proposes Continuously trAnsportable Domain Adaptation (CADA), a framework to handle continuous spurious shifts—situations where spurious correlations between features and labels change smoothly with a continuous domain index (e.g., patient age). The authors develop a causal encoder–discriminator architecture that learns causally transportable representations invariant to such shifts by jointly performing causal inference and adversarial alignment across domains. Theoretical analysis proves CADA ensures causal transportability and bounds target-domain error under covariate shift. Experiments on semi-synthetic (Continuous Colored-MNIST) and real-world medical datasets (SHHS, MESA) demonstrate that CADA outperforms state-of-the-art domain adaptation and causal generalization baselines.

**Strengths:**

- The paper addresses an meaningful problem, domain adaptation under continuous domain indices such as age or time, which is relevant in medical and temporal applications.

**Weaknesses:**

- The motivation is not strongly justified. Although the paper claims to address continuous domain adaptation, the continuous component mainly comes from the discriminator design in CIDA rather than the proposed method itself.

- Some main ideas are theoretically questionable. The authors propose to estimate the $ P(y|do(x)) $. However, the estimation of $ P(y|do(x)) $ is not guaranteed to yield the optimal predictor on observational data, where the optimal hypothesis is Bayes classifier $ P(y|x) $.

- The claim that the method learns causal representations $ v $ is inconsistent with the invariant constraint imposed during training because $ v $ may not invariant across domains; enforcing invariance may alter or remove the causal meaning of $ v $.

- The technical novelty is limited. The proposed framework mainly combines causal representation learning (i.e., front-door adjustment, causal encoder) with standard adversarial alignment similar to CIDA.

- The theoretical results are mostly straightforward restatements of known transportability properties and do not offer new insights. In addition, architectural choices (e.g., Gaussian assumptions, one-sample estimation for $ Q(v|x) )$ lack justification in terms of modeling rationale or computational efficiency.

- The experiments are limited to three datasets (C2MNIST, SHHS, MESA), all in constrained settings. There is no validation on diverse domains such as computer vision, time-series, or NLP, reducing the generalizability of the conclusions. The authors are recommended to experiment with more datasets from literature [1,2]

- The literature review is not comprehensive and omits several recent baselines in continuous or temporal domain adaptation and domain generalization [1,2,3,4,5,6,7].

- The ablation experiments are basic and do not analyze parameter sensitivity, training stability, or robustness under varying domain gaps, leaving many empirical aspects unexplored.

- Code is not provided for reproducibility purposes.

References

[1] Qin, Tiexin, Shiqi Wang, and Haoliang Li. "Generalizing to evolving domains with latent structure-aware sequential autoencoder." International Conference on Machine Learning. 2022.

[2] Pham, Thai-Hoang, Xueru Zhang, and Ping Zhang. "Non-stationary domain generalization: Theory and algorithm." Conference on Uncertainty in Artificial Intelligence. 2024.

[3] Bai, Guangji, Chen Ling, and Liang Zhao. "Temporal Domain Generalization with Drift-Aware Dynamic Neural Networks." International Conference on Learning Representations. 2023.

[4] Zeng, Qiuhao, et al. "Generalizing across temporal domains with koopman operators." AAAI Conference on Artificial Intelligence. 2024.

[5] Zeng, Qiuhao, et al. "Foresee what you will learn: data augmentation for domain generalization in non-stationary environment". AAAI Conference on Artificial Intelligence. 2023.

[6] Xie, Mixue, et al. "Evolving standardization for continual domain generalization over temporal drift." Advances in Neural Information Processing Systems. 2023.

[7] Cai, Zekun, et al. "Continuous temporal domain generalization." Advances in Neural Information Processing Systems. 2024.

**Questions:**

- Could the authors clarify what part of the proposed framework specifically handles the continuous domain index beyond reusing the discriminator from CIDA? Is there any novel mechanism in CADA that models or leverages the continuity of the domain variable (e.g., smoothness constraints, interpolation, or continuous embeddings)?

- Could the authors justify why optimizing $ P(y|do(x)) $ should lead to better generalization performance, and under what assumptions this holds?

- Can the authors provide theoretical or empirical evidence that the learned $v$ preserves causal semantics rather than being just domain-invariant?

- The proposed approach seems to combine causal front-door adjustment with adversarial alignment similar to existing works. Could the authors clarify the unique contribution of CADA beyond integrating these two known ideas? For example, is there any novel insight in how the causal encoder and discriminator interact?

- Can the authors explain why CADA improves over other methods? For instance, do the learned representations show lower mutual information with domain indices or stronger causal disentanglement? Providing quantitative causal metrics or visualizations could strengthen the empirical justification.

---

### Note · Authors · 2025-11-14

I have read and agree with the venue's withdrawal policy on behalf of myself and my co-authors.